# Combination of Cisplatin and Irradiation Induces Immunogenic Cell Death and Potentiates Postirradiation Anti–PD-1 Treatment Efficacy in Urothelial Carcinoma

**DOI:** 10.3390/ijms22020535

**Published:** 2021-01-07

**Authors:** Hiroshi Fukushima, Soichiro Yoshida, Toshiki Kijima, Yuki Nakamura, Shohei Fukuda, Sho Uehara, Yosuke Yasuda, Hajime Tanaka, Minato Yokoyama, Yoh Matsuoka, Yasuhisa Fujii

**Affiliations:** Department of Urology, Tokyo Medical and Dental University, 1-5-45 Yushima, Bunkyo-ku, Tokyo 113-8519, Japan; s-yoshida.uro@tmd.ac.jp (S.Y.); tkijima@dokkyomed.ac.jp (T.K.); nakamura.uro@tmd.ac.jp (Y.N.); shouro@tmd.ac.jp (S.F.); uehauro@tmd.ac.jp (S.U.); yasuuro@tmd.ac.jp (Y.Y.); hjtauro@tmd.ac.jp (H.T.); mntykym.uro@tmd.ac.jp (M.Y.); yoh-m.uro@tmd.ac.jp (Y.M.); y-fujii.uro@tmd.ac.jp (Y.F.)

**Keywords:** chemoradiotherapy, cisplatin, immunotherapy, carcinoma, transitional cell, immunogenic cell death

## Abstract

The therapeutic benefit of immune checkpoint inhibitor monotherapy is limited to a subset of patients in urothelial carcinoma (UC). Previous studies showed the immunogenicity of cisplatin and irradiation. Here, we investigated whether chemoradiotherapy (CRT), a combination of cisplatin and irradiation, could improve the efficacy of postirradiation anti–programmed cell death 1 (PD-1) treatment in UC. In our advanced UC patient cohort, patients with CRT showed a significantly better objective response rate (75%/22%) and overall survival (88%/30% at 12 months) following later pembrolizumab therapy compared to those without. Then, we created syngeneic UC mouse models by inoculating MB49 cells s.c. in C57BL/6J mice to examine the potential of CRT to enhance antitumor immunity in conjunction with postirradiation anti–PD-1 treatment. Nonirradiated tumors of the mice treated with CRT/postirradiation anti–PD-1 treatment had a significantly slower growth rate and a significantly higher expression of cytotoxic T cells compared to those of the mice treated with anti–PD-1 treatment alone. The mice treated with CRT/postirradiation anti–PD-1 treatment showed the best survival. Mechanistically, CRT provoked strong direct cytotoxicity and increased expressions of immunogenic cell death markers in MB49 cells. Therefore, the combination of cisplatin and irradiation induces immunogenic cell death and potentiates postirradiation anti–PD-1 treatment efficacy in UC.

## 1. Introduction

Urothelial carcinoma (UC) encompasses carcinomas of the bladder, ureter, renal pelvis, and urethra. The prognosis of UC patients with metastatic or unresectable locally advanced/recurrent disease is poor, despite initial favorable responses to induction systemic chemotherapy [1]. Previous studies reported that the median duration of overall survival (OS) was 13 to 16 months [1,2]. Thus, the development of effective treatments to improve the prognosis is an unmet need for this lethal disease.

Immune checkpoint inhibitors (ICIs), by which cytotoxic T cells are activated through the abrogation of inhibitory signaling, are the new standard of care for UC [3,4,5]. However, the objective response rate (ORR) of pembrolizumab, an antibody directed against programmed cell death 1 (PD-1), was only 21% in patients with advanced UC that recurred or progressed after platinum-based chemotherapy, according to the KEYNOTE-045 trial [6]. Similarly, in the IMvigor130 trial, the ORR of atezolizumab, an antibody directed against programmed cell death ligand 1 (PD-L1), was 23% in the first-line setting of metastatic UC [2]. These unsatisfactory outcomes have driven many researchers to identify candidate immunomodulators to convert nonresponders into responders [7,8].

Cisplatin-based chemoradiotherapy (CRT) plays a pivotal role in the curative treatment for UC, especially in bladder preservation therapy for muscle-invasive bladder cancer [9,10,11,12]. Radiotherapy can provoke T cell immune responses in tumor tissues [13]. This leads not only to the local control of irradiated tumors but also to systemic antitumor effects (so-called abscopal effects) in nonirradiated tumors [14]. Several preclinical studies showed that irradiation-induced abscopal effects could be synergistically enhanced in conjunction with ICIs in mouse models of various cancers, including UC [15,16,17,18]. Moreover, recent preclinical studies have reported that cisplatin facilitated abscopal effects in combination with irradiation and ICIs [19,20]. Thus, the combination of cisplatin and irradiation can be a promising immunomodulator. However, its role in enhancing antitumor immune activities in conjunction with ICIs is unclear in UC. In addition, there is a paucity of preclinical evidence for the efficacy of postirradiation ICI treatment following the combination of cisplatin and irradiation.

The aim of this study was to investigate whether the combination of cisplatin and irradiation could improve the efficacy of postirradiation anti–PD-1 treatment in UC. First, we evaluated the association of cisplatin-based CRT with the therapeutic outcomes of later pembrolizumab therapy in advanced UC patients. Second, we examined the potential of the combination of cisplatin and irradiation to enhance antitumor immunity in conjunction with postirradiation anti–PD-1 treatment using UC mouse models. We further analyzed the potential of the combination of cisplatin and irradiation to provoke immunogenic cell death, a key mediator of cytotoxic T cell activation and abscopal effects.

## 2. Results

### 2.1. Cisplatin-Based CRT Improved the Efficacy of Later Pembrolizumab Therapy in Patients with Advanced UC

Table 1 shows the patient characteristics of advanced UC patients. The median (range) age was 74 (36–87) years. The primary site was the bladder and upper urinary tract in 18 (69%) and 8 (31%) patients, respectively; 19 (73%) and 15 (58%) patients had lymph node and visceral metastasis, respectively; 8 (31%) and 18 (69%) patients were categorized into the CRT and Non-CRT groups, respectively. All patients in the CRT group were male. The primary site was the bladder in all patients in the CRT group.

During the median (range) follow-up of 6.6 (1.5–27.8) months, 20 (77%)/14 (54%) patients progressed/died. In the total patients, ORR was 38%, and the progression-free survival (PFS) and OS rates at 12-month were 37% and 51%, respectively. The CRT group had a significantly higher ORR than the Non-CRT group (75% vs. 22%; Figure 1A). Although statistical significance was not reached, the CRT group showed a higher PFS rate than the Non-CRT group (63% vs. 27% at 12 months; Figure 1B). The CRT group showed a significantly higher OS rate compared to the Non-CRT group (88% vs. 30% at 12 months; Figure 1C). In addition, when analyzed only in 18 bladder cancer patients, the OS rate was significantly higher in the CRT group than in the Non-CRT group (88% vs. 20% at 12 months; Figure 1D). Thus, our clinical data indicated that cisplatin-based CRT could improve the efficacy of later pembrolizumab therapy in patients with advanced UC.

### 2.2. Abscopal Effects in a UC Mouse Model

A previous study showed that hypofractionated radiotherapy induced abscopal effects in conjunction with concurrent anti–PD-L1 treatment using C57BL/6 mice inoculated with murine UC cell line MB49 [15]. Based on this previous work, we conducted a pilot study of the MB49 model in which left hindlimb tumors were irradiated (right flank tumors were not irradiated) and either anti–PD-1 antibody or isotype control was administered (Figure 2A) to evaluate whether abscopal effects could be induced by a single dose of irradiation in conjunction with concurrent anti–PD-1 treatment in a UC mouse model. There was no significant difference of the growth rates of irradiated left hindlimb tumors between the mice treated with irradiation alone and the control mice (Figure 2B, left), reflecting poor antitumor effects of irradiation alone. The mice treated with irradiation/anti–PD-1 treatment showed the best outcome (Figure 2B). The growth rate of nonirradiated right flank tumors in the mice treated with irradiation/anti–PD-1 treatment was significantly slower than that in the mice treated with anti–PD-1 treatment alone, reflecting irradiation-induced abscopal effects (Figure 2B, right). Moreover, the combination of irradiation/anti–PD-1 treatment resulted in the best survival (Figure 2C). Against this background, we used the MB49 model for further investigation.

### 2.3. Combination of Cisplatin and Irradiation Promoted Antitumor Effects of Later Anti–PD-1 Treatment in UC

Next, we created CRT and Non-CRT models to evaluate the potential of the combination of cisplatin and irradiation to enhance antitumor immune activities in conjunction with later anti–PD-1 treatment. In the CRT model, left hindlimb tumors were irradiated in conjunction with cisplatin treatment (CRT), followed by the development of right flank tumors. Afterward, either anti–PD-1 antibody or isotype control was administered (Figure 3A, upper). In the Non-CRT model, after only right flank tumors were developed, either anti–PD-1 antibody or isotype control was administered (Figure 3A, lower). As for irradiated left hindlimb tumors, the mice treated with CRT/postirradiation anti–PD-1 treatment had a significantly better response than those treated with CRT alone (Figure 3B, left and Appendix A, left). Nonirradiated right flank tumors in the mice treated with CRT/postirradiation anti–PD-1 treatment had a significantly slower growth rate compared to those treated with anti–PD-1 treatment alone (Figure 3B, right and Appendix A, right). In addition, the group treated with CRT/postirradiation anti–PD-1 treatment showed the best survival (Figure 3C).

To investigate the effect of the combination of cisplatin and irradiation on the expression of cytotoxic T cells in conjunction with postirradiation anti–PD-1 treatment, the immune cell expression profile was compared using the nonirradiated right flank tumors of the mice treated with anti–PD-1 treatment alone and both the irradiated left hindlimb and nonirradiated right flank tumors of the mice treated with CRT/postirradiation anti–PD-1 treatment (Figure 3D–G). The expression levels of CD8^+^CD4^−^ T cells were similar among the three tumors (Figure 3F). However, the expression levels of IFN-γ^+^CD8^+^CD4^−^ T cells were significantly higher in the irradiated left hindlimb and nonirradiated right flank tumors of the mice treated with CRT/postirradiation anti–PD-1 treatment compared to the nonirradiated right flank tumors of the mice treated with anti–PD-1 treatment alone (Figure 3G). IFN-γ^+^CD8^+^CD4^−^ T cell expression did not differ between the irradiated left hindlimb and nonirradiated right flank tumors of the mice treated with CRT/postirradiation anti–PD-1 treatment (Figure 3G), suggesting systemic antitumor immunity induced by CRT and postirradiation anti–PD-1 treatment. Therefore, the combination of cisplatin and irradiation, in conjunction with postirradiation anti–PD-1 treatment, elicited antitumor effects in nonirradiated left hindlimb tumors, in which the expression of cytotoxic T cells was increased in the tumor microenvironment.

### 2.4. Combination of Cisplatin and Irradiation Strongly Exerted Direct Cytotoxic Effects on UC

The cytotoxic effects of cisplatin on MB49 cells were evaluated in vitro. MB49 cells were incubated with each concentration of cisplatin in vitro, and cell viability was reduced in a dose-dependent manner (Appendix A). According to the results of the cell viability assay, the half-maximal inhibitory concentration (IC50) of cisplatin was 0.66 mg/dL. Thus, we used a concentration of 0.6 mg/dL for cisplatin in subsequent in vitro experiments.

Next, we evaluated the direct cytotoxic effects of the combination of cisplatin and irradiation on MB49 cells. The proportion of dead cells two days after in vitro treatment was the highest in the group treated with CRT, though there was no significant difference between the group treated with cisplatin alone and the group treated with CRT (Appendix A). This was confirmed by an in vivo analysis; the tumor growth rate of the group treated with CRT was significantly slower compared to that of the other groups (Appendix A). Thus, the combination of cisplatin and irradiation strongly exerted direct cytotoxic effects on UC.

### 2.5. Combination of Cisplatin and Irradiation Strongly Induced Immunogenic Cell Death in UC

Immunogenic cell death of tumor cells is a key to activating antitumor cytotoxic T cells and harnessing abscopal effects [21,22]. Thus, we examined whether the combination of cisplatin and irradiation can induce immunogenic cell death in UC. Expressions of high mobility group box 1 (HMGB1) and calreticulin proteins, damage-associated molecular patterns (DAMPs) characterizing immunogenic cell death, were analyzed after in vitro treatment. Secreted HMGB1 levels were highest in the group treated with CRT (Figure 4A). Similarly, calreticulin expressions on the tumor cell surface were highest in the group treated with CRT (Figure 4B). The induction of immunogenic cell death by CRT was also evaluated in vivo. Immunohistochemistry analysis revealed that HMGB1 and calreticulin expression levels were significantly higher after CRT than before CRT (Figure 4C, right and Figure 4D, right). As shown in the representative images (Figure 4C, left), HMGB1 was positive mainly at the nucleus before treatment, whereas HMGB1 staining at the cytoplasm or cell membrane was also observed after treatment, suggesting the increased secretion of the nuclear protein HMGB1 after CRT.

## 3. Discussion

In the present study, we investigated whether the combination of cisplatin and irradiation could improve the efficacy of postirradiation anti–PD-1 treatment in UC. In our cohort of advanced UC patients, the CRT group had a significantly higher ORR and a significantly improved OS following later pembrolizumab therapy compared to the Non-CRT group. Moreover, we revealed in vivo that the combination of cisplatin and irradiation, in conjunction with postirradiation anti–PD-1 treatment, exerted antitumor effects in nonirradiated tumors, in which an increase in cytotoxic T cells was observed in the tumor microenvironment. These results reflect that postirradiation anti–PD-1 treatment can enhance abscopal effects induced by the combination of cisplatin and irradiation. Mechanistically, the combination of cisplatin and irradiation strongly induced direct cytotoxicity and immunogenic cell death in UC. Because the immunogenic cell death of tumor cells results in the activation of antitumor cytotoxic T cells [21,22], the combination of cisplatin and irradiation can potentiate the efficacy of postirradiation anti–PD-1 treatment by inducing immunogenic cell death in UC (refer to graphical abstract). Because irradiation upregulates PD-1 expression in CD8^+^ T cells in irradiated tumors [17,23], anti–PD-1 antibody can enhance the antitumor immune activities of PD-1 expressing immune-primed T cells that migrate to nonirradiated tumors. Given that cisplatin-based CRT is an established therapy in UC, including bladder preservation therapy for muscle-invasive bladder cancer [9,10,11,12], our findings suggest that cisplatin-based CRT and postirradiation anti–PD-1 treatment may be a clinically viable and promising therapeutic strategy in UC patients.

Numerous studies have verified the immunogenic role of irradiation [14,15,16,17,18]. Irradiation increases T cell receptor diversity by increasing tumor-associated antigen release [24] and interferon secretion through the cyclic GMP-AMP synthase (cGAS)-stimulator of interferon genes (STING) pathway [25]. In addition, irradiation induces the expression and release of DAMPs such as calreticulin and HMGB1 that characterize immunogenic cell death, resulting in the activation of cytotoxic T cells [13]. The immunogenic potential of cisplatin is controversial. Several recent studies have shown that cisplatin provoked immunogenic cell death to the same extent as oxaliplatin, which is a well-known immunogenic cell death inducer [26]. Thus, cisplatin is potentially immunogenic. In our in vitro analysis, the combination of cisplatin and irradiation induced immunogenic cell death to a greater extent compared to each monotherapy, suggesting the strong immunogenic potential of the combination of cisplatin and irradiation. Moreover, because cisplatin is a radiosensitizer [27], the combination of cisplatin and irradiation synergistically eradicates tumor cells, which may increase immune priming.

The optimal sequencing of irradiation and ICIs is controversial. Dovedi et al. reported that mice treated with anti–PD-L1 treatment and concurrent irradiation had longer survivals compared to those treated with anti–PD-L1 treatment seven days after irradiation in vivo [28]. However, the concurrent use strategy may not be clinically feasible because previous clinical trials were discontinued due to unacceptably high rates of severe adverse events following the concurrent use of hypofractionated radiotherapy and anti–PD-1/PD-L1 treatment in UC patients [29,30]. Meanwhile, recent findings from clinical data, including ours, showed that a history of radiotherapy was significantly associated with favorable prognosis after pembrolizumab therapy in patients with advanced UC [31] and non-small cell lung cancer [32], in which a history of radiotherapy was not associated with higher rates of adverse events. Thus, postirradiation anti–PD-1 treatment may safely enhance irradiation-induced abscopal effects. In our in vivo analysis, we revealed that the combination of cisplatin and irradiation improved the efficacy of postirradiation anti–PD-1 treatment, which was initiated 21 days after the combination of cisplatin and irradiation. Our results suggest that immunological memory may mediate antitumor effects in nonirradiated tumors after postirradiation anti–PD-1 treatment. Further research is necessary to clarify the effect of the combination of cisplatin and irradiation on immunological memory.

Several limitations exist in the present study that should be addressed. First, our clinical results were subject to selection biases due to the small sample size of our advanced UC cohort. Second, we did not investigate the direct mechanism by which immunogenic cell death activates antitumor immune responses. This has, however, been previously demonstrated in numerous studies [21,22]. Immunogenic cell death induces the secretion of various chemokines, resulting in the activation of cytotoxic T cells. In the setting of the combination of cisplatin and irradiation, C-X-C motif chemokine ligand 10 (CXCL10), a T cell chemoattractant, mediated antitumor effects in conjunction with concurrent anti–PD-1 treatment [20]. It should be further elucidated how chemokines work in antitumor immune activities induced by the combination of cisplatin and irradiation. Third, we used a single dose of irradiation. Previous studies reported the superiority of multi-fractionated regimens in inducing abscopal effects [33]. Thus, we confirmed the potential of a single dose of irradiation to induce abscopal effects in conjunction with anti–PD-1 treatment (Figure 2). Fourth, changes in PD-1/PD-L1 expressions following CRT were not analyzed in the present study. Finally, the role of immunosuppressive cells in abscopal effects by the combination of cisplatin and irradiation was not evaluated in the present study. We will examine this issue in future research.

## 4. Materials and Methods

### 4.1. Clinical Investigations in an Advanced UC Patient Cohort

We retrospectively reviewed 26 consecutive advanced UC patients (lymph node/visceral metastasis or unresectable locally advanced/recurrent disease) treated with pembrolizumab as a second or later line therapy between January 2018 and December 2019 in a single institution. All patients had pathologically confirmed urothelial carcinoma. A fixed dose of 200 mg/body pembrolizumab was intravenously infused every three weeks. Eight (31%) patients who received cisplatin-based CRT to the primary site within the past two years of the initiation of pembrolizumab therapy were categorized into the CRT group, and the remaining 18 (69%) patients were assigned to the Non-CRT group. Cisplatin-based CRT consisted of external beam radiotherapy with 40 Gy in 20 fractions (4-field box irradiation using 10 MV X-ray) and two cycles of concurrent intravenous infusion of cisplatin (20 mg/d for 5 days) with a cisplatin-free interval of two weeks [11,12]. The cisplatin dose was reduced according to each patient’s general condition and renal function. ORR was defined as the percentage of patients with complete and partial responses based on the best overall response at the last follow-up, which was evaluated by institutional radiologists based on the Response Evaluation Criteria in Solid Tumors (RECIST), version 1.1 [34]. PFS was calculated from the initiation of pembrolizumab therapy to disease progression, death, or last follow-up. OS was calculated from the initiation of pembrolizumab therapy to death or last follow-up.

### 4.2. Mice and Cell Line

C57BL/6J five- to six-week-old female mice were obtained from CLEA Japan (Tokyo, Japan) and maintained under pathogen-free conditions in the animal facility of the Tokyo Medical and Dental University. MB49, a mouse bladder UC cell line, was purchased from Merck KGaA (Darmstadt, Germany). It was cultured in RPMI1640 supplemented with 10% fetal bovine serum and 1% penicillin/streptomycin at 37 °C and 5% CO_2_. Cells were confirmed to be free of mycoplasma contamination using a MycoAlert Mycoplasma Detection Kit (Lonza, Basel, Switzerland). Cell count was conducted using a TC20 automated cell counter (Bio-Rad, Hercules, CA, USA).

### 4.3. In Vivo Tumor Inoculation and Treatment

To assess the abscopal effects of a single dose of irradiation in mouse models, six- to seven-week-old C57BL/6J mice were subcutaneously inoculated with MB49 cells (2 × 10^6^, in Matrigel) in the left hindlimb and the right flank, respectively. Seven days later, they received irradiation with a single fraction of 10 Gy to the left hindlimb and anti–PD-1 treatment (or its isotype controls). To analyze the potential of the combination of cisplatin and irradiation to enhance antitumor immune activities in conjunction with later anti–PD-1 treatment in mouse models, we created CRT and Non-CRT models. In the CRT model, six- to seven-week-old C57BL/6J mice were injected with MB49 cells (5 × 10^4^) in the left hindlimb, and seven days later, received cisplatin and irradiation with a single fraction of 10 Gy to the left hindlimb. The mice were injected with MB49 cells (2 × 10^6^, in Matrigel) in the right flank 14 days after irradiation, and seven days later, received anti–PD-1 treatment (or its isotype controls). For the Non-CRT model, MB49 cells (2 × 10^6^, in Matrigel) were injected only in the right flank, and seven days later, received anti–PD-1 treatment (or its isotype controls). Tumor size was measured with calipers twice a week, and tumor volumes were calculated using the formula: (length^2^ × width)/2. In the CRT model, when left hindlimb tumors reached 20 to 50 mm^3^ in size, irradiation was delivered. In the CRT and Non-CRT models, when right flank tumors reached at least 20 mm^3^ in size, anti–PD-1 treatment (or its isotype controls) was initiated. Mice were euthanized when either primary or secondary tumors reached 2000 mm^3^ in size.

Anti–PD-1 antibody (clone RMP1-14, Leinco Technologies, Inc., St. Louis, MO, USA) or isotype control antibody (clone 1-1, Leinco Technologies, Inc., St. Louis, MO, USA) were intraperitoneally injected at 10 mg/kg twice per week for four weeks. Cisplatin (Merck KGaA, Darmstadt, Germany) was intraperitoneally injected at 3 mg/kg on the day of irradiation.

### 4.4. Tumor Irradiation

A single fraction of 10 Gy irradiation was delivered to mice or tumor cells using an X-ray generator (MBR-1520R-4; Hitachi, Tokyo, Japan) at a dose rate of 0.57 Gy/min. Mice were anesthetized, positioned in a custom-made plastic container, and shielded with lead to allow irradiation to the left hindlimb and to protect the rest of the body.

### 4.5. Flow Cytometric Analysis

Mice were euthanized with CO_2_ inhalation and subcutaneous tumors were excised. Tumor tissues were dissociated using BD Horizon Dri Tumor & Tissue Reagent (BD Biosciences, Franklin Lakes, NJ, USA) to obtain single-cell suspensions. Fluorochrome-conjugated antibodies used for flow cytometric analysis were obtained from BD Biosciences (Franklin Lakes, NJ, USA). To analyze tumor-infiltrating lymphocytes, cells were stained with PE-Cy7-conjugated CD3e (clone 145-2C11), APC-conjugated CD4 (RM4-5), APC-Cy7-conjugated CD8a (53-6.7), BV510-conjugated CD45 (30-F11), and PE-conjugated IFN-γ (XMG 1.2) antibodies. Anti-CD16/CD32 antibody (BD Biosciences) was used to block Fc receptors. 7-AAD (BD Biosciences, Franklin Lakes, NJ, USA) was used to select viable cells. For the intracellular staining of IFN-γ, cells were stimulated by PMA (1:10,000) and ionomycin (1:2000) in the presence of GolgiPlug at 37 °C for 4 h and processed using a Fixation/Permeabilization Solution Kit (BD Biosciences, Franklin Lakes, NJ, USA) according to the manufacturer’s protocol. After extracellular staining, cells were fixed and permeabilized, followed by intracellular staining with IFN-γ.

MB49 cells were seeded at 2 × 10^6^ cells in 10 cm plates and incubated overnight to allow cell attachment prior to in vitro experiments. Cells were exposed to cisplatin (0.6 mg/L) and/or irradiated with a single dose of 10 Gy. After 24 h of incubation, the medium was replaced with fresh serum-free medium. To analyze the cell surface expression of calreticulin, cells were suspended in phosphate-buffered saline and stained with ALEXA FLUOR 488 conjugated anti-calreticulin polyclonal antibody (Bioss Inc., Woburn, MA, USA), followed by 7-AAD (BD Biosciences, Franklin Lakes, NJ, USA) staining to exclude dead cells. To evaluate the cytotoxicity of irradiation and cisplatin, cells were suspended in phosphate-buffered saline and stained with PE-conjugated anti-Annexin V (BD Biosciences, Franklin Lakes, NJ, USA) and 7-AAD.

All samples were run on a BD FACSCanto II flow cytometer system (BD Biosciences, Franklin Lakes, NJ, USA). Data analysis was performed using FlowJo v10 software.

### 4.6. Enzyme-Linked Immunoassay (ELISA)

HMGB1 secretion from MB49 cells was assessed in vitro by ELISA. MB49 cells were seeded at 2 × 10^6^ cells in 10 cm plates and incubated overnight to allow cell attachment prior to treatment. Cells were exposed to cisplatin (0.6 mg/L) and/or irradiated with a single dose of 10 Gy. After 24 h of incubation, the medium was replaced with fresh serum-free medium. Fresh supernatants were collected and assessed using an ELISA kit for HMGB1 according to the manufacturer’s instructions (Arigo Biolaboratories Corp., Hsinchu, Taiwan). Plates were analyzed with a FLUOstar OPTIMA microplate reader (BMG LABTECH, Offenbrug, Germany).

### 4.7. Cytotoxicity Assay

Cisplatin cytotoxicity and the IC50 of cisplatin were analyzed in vitro using a CellTiter 96 AQueous One Solution Cell Proliferation Assay kit (Promega KK, Madison, WI, USA) according to the manufacturer’s instructions. MB49 cells were seeded at 5 × 10^3^ cells/well in a 96-well plate and incubated overnight to allow cell attachment. Cells were cultured in a medium with cisplatin at each concentration (0, 0.25, 0.4, 0.5, 1, 2.5, 5, 10 mg/L) for 24 h, and then the medium was replaced with 100 μL of fresh medium. After 24 h of incubation, 20 μL of MTS solution was added to each well. After 1 h of incubation, plates were analyzed using a FLUOstar OPTIMA microplate reader (BMG LABTECH, Offenbrug, Germany). Six wells were assayed for each cisplatin concentration, and the mean absorbance was calculated.

### 4.8. Immunohistochemistry

To analyze HMGB1 and calreticulin protein expressions in mouse tumor tissues, six- to seven-week-old mice were injected with MB49 cells (2 × 10^6^) in the left hindlimb, and seven days later, treated with cisplatin at 3 mg/kg and irradiation with a single fraction of 10 Gy to the left hindlimb. Mice were euthanized immediately before treatment and seven or 28 days after treatment, and subcutaneous tumors were excised. Tissue sections were deparaffinized with xylene and rehydrated through an ethanol series and phosphate-buffered saline. Antigen retrieval was performed by microwave treatment with Tris-EDTA buffer (pH 9.0). Endogenous peroxidase was blocked with 0.3% H_2_O_2_ in methanol for 30 min, followed by incubation with G-Block (Genostaff, Tokyo, Japan) and an avidin/biotin blocking kit (Vector, Burlingame, CA, USA). The sections were incubated with anti-HMGB1 rabbit monoclonal antibody (EPR3507; Abcam, Cambridge, UK) or anti-calreticulin rabbit polyclonal antibody (Bioss Inc., Woburn, MA, USA) at 4 °C overnight. They were then incubated with biotin-conjugated anti-rabbit Ig (Agilent, Santa Clara, CA, USA) for 30 min at room temperature, followed by the addition of peroxidase-conjugated streptavidin (Nichirei, Tokyo, Japan) for 5 min. Peroxidase activity was visualized by diaminobenzidine. The sections were counterstained with Mayer’s Hematoxylin (MUTO, Tokyo, Japan), dehydrated, and then mounted with Malinol (MUTO, Tokyo, Japan). The number of cells with membranous or cytoplasmic expressions of HMGB1 and calreticulin were counted in five randomly selected fields of view (FOV) according to the previous study. Normal mouse liver and brain tissues were used as positive controls for HMGB1 and calreticulin expressions, respectively. In negative controls, the primary antibody was substituted for normal rabbit Ig.

### 4.9. Statistical Analysis

Quantitative data were presented as the mean plus or minus the standard error of the mean. The chi-square or Fisher’s exact test was used to compare categorical variables, and Student’s or Welch’s *t*-test or Mann–Whitney U test was used to compare the mean values of continuous variables. Survival curves were depicted by the Kaplan–Meier method, and their differences were assessed by the log-rank test. Statistical analyses were performed using JMP 9.0.2 (SAS Institute Inc., Cary, NC, USA). A two-tailed *p* < 0.05 was considered statistically significant.

## 5. Conclusions

The combination of cisplatin and irradiation induces immunogenic cell death and potentiates the efficacy of postirradiation anti–PD-1 treatment in UC. Therefore, cisplatin-based CRT and postirradiation anti–PD-1 treatment may be a clinically viable and promising therapeutic strategy in UC.

## Figures and Tables

**Figure 1 ijms-22-00535-f001:**
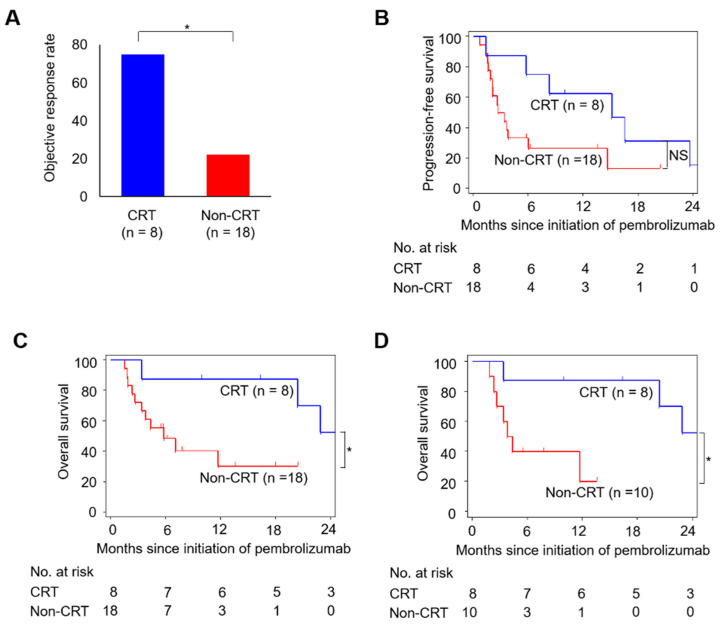
Associations of cisplatin-based CRT with therapeutic response and survival after pembrolizumab therapy in patients with advanced urothelial carcinoma (UC). (**A**) Objective response rate (ORR), (**B**) progression-free survival (PFS), and (**C**) overall survival (OS) are compared between CRT (*n* = 8) and Non-CRT (*n* = 18) groups. (**D**) Comparison of OS curves between CRT (*n* = 8) and Non-CRT (*n* = 10) groups in 18 bladder cancer patients. The *p*-value is calculated by the chi-square test for ORR and the log-rank test for PFS and OS. Asterisks indicate *p*-values comparing two groups, as indicated in the figure. NS, not significant; * *p* < 0.05.

**Figure 2 ijms-22-00535-f002:**
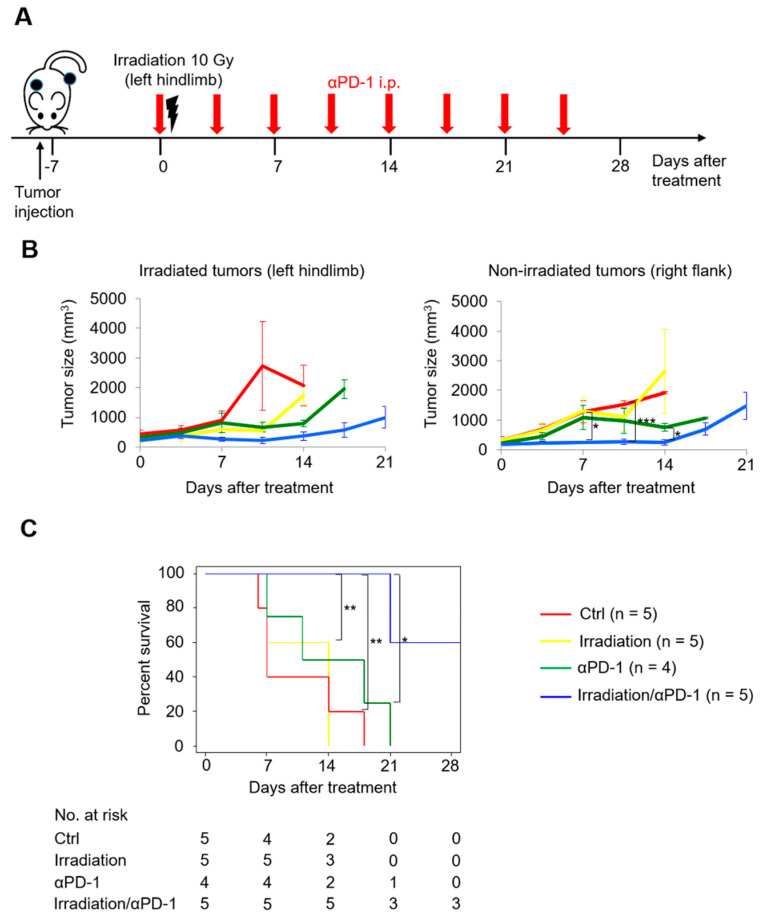
Concurrent anti–PD-1 treatment facilitated abscopal effects induced by a single dose of irradiation in MB49 tumor-bearing mice. (**A**) Scheme for tumor inoculation and treatments. Six- to seven-week-old mice are inoculated subcutaneously with MB49 cells in the left hindlimb and in the right flank, and seven days later, receive irradiation with a single fraction of 10 Gy to the left hindlimb and anti–PD-1 treatment (or its isotype controls). (**B**) MB49 tumor growth curves of irradiated tumors in the left hindlimb (left) and nonirradiated tumors in the right flank (right). (**C**) Survival curves of mice. Log-rank test is used to compare survival curves. All data are shown as the mean ± standard error of the mean (SEM). Asterisks indicate *p*-values comparing two groups, as indicated in the figure. * *p* < 0.05; ** *p* < 0.01; *** *p* < 0.001.

**Figure 3 ijms-22-00535-f003:**
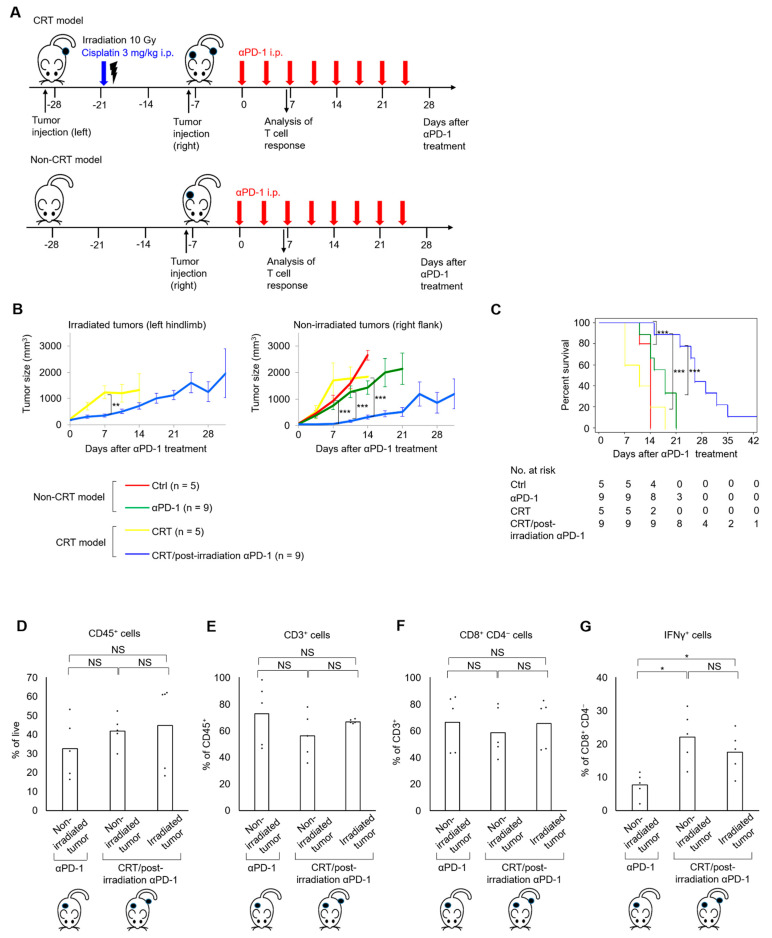
The combination of cisplatin and irradiation potentiated the efficacy of postirradiation anti–PD-1 treatment in MB49 tumor-bearing mice. (**A**) Scheme for tumor inoculation, treatments, and T cell analysis. For the CRT model (top), six- to seven-week-old mice are injected with MB49 cells in the left hindlimb, and seven days later, receive cisplatin at 3 mg/kg and irradiation with a single fraction of 10 Gy to the left hindlimb. The mice are injected with MB49 cells in the right flank 14 days after irradiation, and seven days later, receive anti–PD-1 treatment (or its isotype controls). For the Non-CRT model (bottom), only a right flank tumor is established in the mice, and cisplatin and irradiation are not given. (**B**) MB49 tumor growth curves of irradiated tumors in the left hindlimb (left) and nonirradiated tumors in the right flank (right). (**C**) Survival curves of mice. The log-rank test is used to compare survival curves. (**D**–**G**) Proportions of CD45^+^ cells in the live cells (**D**), CD3^+^ cells in the CD45^+^ subpopulation (**E**), CD8^+^CD4^−^ cells in the CD3^+^ subpopulation (**F**), and IFNγ^+^ cells in the CD8^+^CD4^−^ subpopulation (**G**) in single-cell suspensions of the irradiated and nonirradiated tumors on day seven in mice treated with CRT/postirradiation anti–PD-1 treatment compared to those of the nonirradiated tumors on day seven in mice treated with anti–PD-1 treatment alone (*n* = 5/group). All data are shown as the mean ± standard error of the mean (SEM). Asterisks indicate *p*-values comparing two groups, as indicated in the figure. NS, not significant; * *p* < 0.05; ** *p* < 0.01; *** *p* < 0.001.

**Figure 4 ijms-22-00535-f004:**
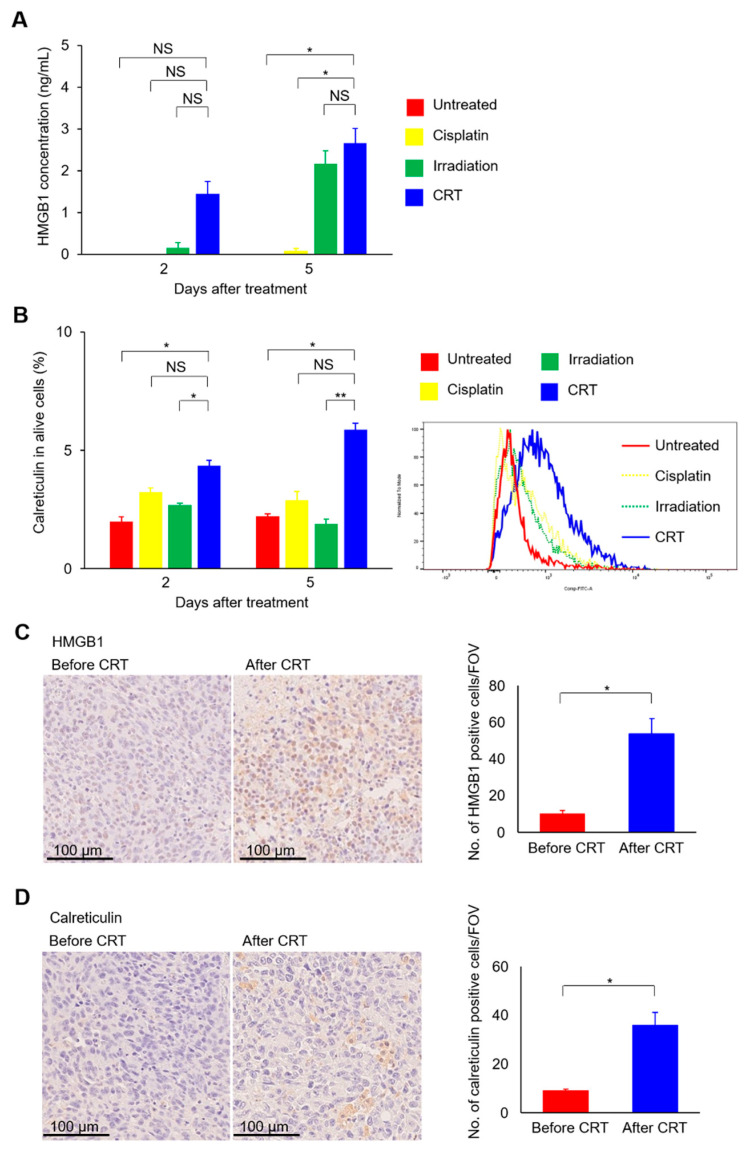
The combination of cisplatin and irradiation increases HMGB1 protein secretion and cell surface expression of calreticulin protein in MB49 cells. (**A**) MB49 cells are treated with cisplatin at 0.6 mg/L and/or irradiated with a single fraction of 10 Gy. Medium is collected two and five days after treatment, and HMGB1 protein expression levels in each medium are examined in duplicate by ELISA (*n* = 3/group). (**B**) MB49 cells are treated with cisplatin at 0.6 mg/L and/or irradiated with a single fraction of 10 Gy. Cells are collected, and cell surface calreticulin protein expression levels are examined by flow cytometry (*n* = 3/group). (**C**,**D**) Seven days after inoculation MB49 cells in the left hindlimb, mice are treated with cisplatin at 3 mg/kg and/or irradiation with a single fraction of 10 Gy. Tumor sections before and after CRT (*n* = 3/group) are stained for HMGB1 (**C**) and calreticulin (**D**). Left, representative images; right, comparison of the mean (± standard error of the mean [SEM]) number of HMGB1- or calreticulin-positive cells counted in five fields of view (FOV). All data are shown as the mean ± SEM. Asterisks indicate *p* values comparing two groups, as indicated in the figure. NS, not significant; * *p* < 0.05; ** *p* < 0.01.

**Table 1 ijms-22-00535-t001:** Patient characteristics.

Variables		Overall, *n* (%)	CRT Group, *n* (%)	Non-CRT Group, *n* (%)	*p*-Value
No. of patients		26 (100)	8 (31)	18 (69)	
Age (years)		74 (36–87)	74 (46–86)	76 (36–87)	0.96
Sex	Male	19 (73)	8 (100)	11 (61)	0.039
	Female	7 (27)	0 (0)	7 (39)	
ECOG PS	0–1	23 (88)	7 (88)	16 (89)	0.92
	≥2	3 (12)	1 (13)	2 (11)	
Primary site	Bladder	18 (69)	8 (100)	10 (56)	0.023
	UUT	8 (31)	0 (0)	8 (44)	
Lymph node metastasis	No	7 (21)	1 (13)	6 (33)	0.27
	Yes	19 (73)	7 (88)	12 (67)	
Visceral metastasis	No	11 (42)	2 (25)	9 (50)	0.23
	Yes	15 (58)	6 (75)	9 (50)	
Previous curative surgery	No	15 (42)	5 (63)	10 (56)	0.74
	Yes	11 (58)	3 (38)	8 (44)	
Line of pembrolizumab	2nd	23 (88)	7 (88)	16 (89)	0.92
	3rd or later	3 (12)	1 (13)	2 (11)	

Abbreviations: CRT, chemoradiotherapy; ECOG PS; Eastern Cooperative Oncology Group Performance Status; UUT, upper urinary tract.

## Data Availability

Data available on request due to privacy/ethical restrictions.

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
