# Peer review of "Combination of Cisplatin and Irradiation Induces Immunogenic Cell Death and Potentiates Postirradiation Anti–PD-1 Treatment Efficacy in Urothelial Carcinoma"

_ijms, 2021, doi:10.3390/ijms22020535_

Round 1
Reviewer 1 Report
In the given study, the authors have investigated the efficacy of chemoradiotherapy (CRT) to improve anti-tumor effects of post-irradiation anti-programmed cell death 1 treatment in UC using a syngeneic UC mouse model by inoculating MB49 cells s.c. in C57BL/6J mice. The study addresses an important issue in the field, is well designed, and most part, the results are supportive of the conclusions made by the authors.
I have a few comments that should be addressed to improve the quality of this study:
- How do authors explain the finding that TIL IFNy+ CD8+ T cells donot differ between IR vs Non-IR treated groups. This and no significant differences between overall CD8+ cells should be addressed atleast in discussion.
- Authors should check the levels of PD1, and PDL1 in T cells, and tumor cells respectively, under different conditions.
- Graphical abstract (as indicated in Line 203) , is missing , and should be good to include to summarize the findings.
Author Response
Dear the Reviewer,
We greatly appreciated the reviewer for taking their precious time to review our paper and giving constructive comments. We revised the manuscript according to the comments. Please take a look at the changes we made in the revised version outlined below.
1. How do authors explain the finding that TIL IFNy+ CD8+ T cells do not differ between IR vs Non-IR treated groups. This and no significant differences between overall CD8+ cells should be addressed at least in discussion.
First, we do thank you for all the constructive comments. IFN-γ+CD8+CD4- T cell expression did not differ between the irradiated left hindlimb and non-irradiated right flank tumors of the mice treated with CRT/post-irradiation anti-PD-1 treatment. This suggests systemic anti-tumor immunity induced by CRT and post-irradiation anti-PD-1 treatment.
Thus, we revised sentences in the Results section as follows (line 158-161): IFN-γ+CD8+CD4- T cell expression did not differ between the irradiated left hindlimb and non-irradiated right flank tumors of the mice treated with CRT/post-irradiation anti-PD-1 treatment (Figure 3G), suggesting systemic anti-tumor immunity induced by CRT and post-irradiation anti-PD-1 treatment.
In addition, the expression levels of IFN-γ+CD8+CD4- T cells were significantly higher in the irradiated left hindlimb and non-irradiated right flank tumors of the mice treated with CRT/post-irradiation anti-PD-1 treatment compared to the non-irradiated right flank tumors of the mice treated with anti-PD-1 treatment alone, but CD8+CD4- T cell expression was statistically similar. These phenomena can be explained by assuming that post-irradiation anti-PD-1 treatment could enhance CRT-induced abscopal effects.
Thus, we added the following sentences to the Discussion section (line 221-222): “These results reflect that post-irradiation anti-PD-1 treatment can enhance abscopal effects induced by the combination of cisplatin and irradiation.”
2. Authors should check the levels of PD1, and PDL1 in T cells, and tumor cells respectively, under different conditions.
We appreciate the reviewer’s pointing out. This point is very important in enhancing the efficacy of immune checkpoint inhibitors, but it requires extensive experiments. Thus, we commented this point as a limitation of this study in the Discussion section. In addition, because a previous in vivo analysis by Dovedi et al. showed that PD-1 up-regulation in CD4+CD8+ T cells was observed 1 day after radiotherapy but not 7 days after radiotherapy (Dovedi, S. J. et al. Acquired resistance to fractionated radiotherapy can be overcome by concurrent PD-L1 blockade. Cancer Res 2014, 74, (19), 5458-68.), we suspect that an increase in PD-1 expression induced by radiotherapy may be non-durable. We would start a new project to chronologically evaluate the changes in PD-1/PD-L1 expressions in the tumor microenvironment after CRT.
Thus, we added the following sentence to the Discussion section (line 293-294): “Fourth, changes in PD-1/PD-L1 expressions following CRT were not analyzed in the present study.”
3. Graphical abstract (as indicated in Line 203), is missing, and should be good to include to summarize the findings.
We appreciate the reviewer’s helpful comments. I uploaded the graphical abstract.
Reviewer 2 Report
Overview.
Fukushima et al. demonstrate the potential of considering prior localized radiation and chemotherapies (cis-platin) when treating advanced urothelial cancer patients with anti-PD-1 immune modulatory therapies. This could have clinical utility when decisions are being made for treatments of the metastatic cancers and potentially suggests that radiation and chemo, in addition to having great impacts on the primary tumor, should be discussed if the patients are at high risk for developing advance disease. In general, the major conclusions are supported, and the language is very clear. Each figure supports that the combination therapies look effective based on the clinical or tumor experiments. This is associated with increased cytotoxicity in the tumor cells. The authors also utilized an interesting in vivo model with implantations on both flanks of mice in the experiments, which limits the potential variation from mice to mice to improve the precision of the outcomes. There are some specific areas in which the authors should improve how they communicate the data and rationalize the experimental models. These are discussed below:
Revision comments.
- In the clinical study, it is possible that biology specific to the primary tumor site may dictate some degree of response to combination therapies. Does the responses and survival trends hold true, or become more robust, when only comparing the 8 CRT to the 10 of 18 non-CRT that originate from the bladder cancers? If not, please justify why the inclusions of the 8 UUTs impacted the results.
- Authors suggest a single dose radiation would be sufficient in improving the combo therapy and thus set up two in vivo models to specifically irradiate the tumor in the left and not the right flank of the mice (methods). This is a key procedure, but it is not well depicted in the schematic in Figure 2A. It is more apparent in Figure 3A. As the radiation is site specific, but the other systemic therapies are not accurately depicting the schematics would improve the reader’s understanding of the experiments.
- An additional point regarding these experiments, better description, depictions or presentation of the results in 2B and 3B would also help readers as well. The current labels are “irradiated” and “non-irradiated”; but the yellow and blue lines state they are in the CRT group that received radiation. With the brief legends and description in the results, it is difficult for me to understand if the right panels ever received irradiation. In addition, a direct comparison between the tumors in the left and right flanks of each mice would better illustrate the specific effects of radiation.
- Additional descriptions should be added in the results and potentially methods to justify why experiments were conducted in this particular mouse model/strain (C57BL/6J) and the cancer cell line (MB49). Particularly, emphasis should be placed on how the immune system of this model is suited towards studying urothelial cancer interaction with immune therapies. Additional citations would bolster the credibility of the results derived from these models.
Author Response
Dear the Reviewer,
We greatly appreciated the reviewer for taking their precious time to review our paper and giving constructive comments. We revised the manuscript according to the comments. Please take a look at the changes we made in the revised version outlined below.
1. In the clinical study, it is possible that biology specific to the primary tumor site may dictate some degree of response to combination therapies. Does the responses and survival trends hold true, or become more robust, when only comparing the 8 CRT to the 10 of 18 non-CRT that originate from the bladder cancers? If not, please justify why the inclusions of the 8 UUTs impacted the results.
First, we do thank you for all the constructive comments. When analyzed in 18 bladder cancer patients, the CRT group showed a significantly higher OS rates compared to the Non-CRT group. Thus, we commented in the Results section as follows (line 88-90): “In addition, when analyzed only in 18 bladder cancer patients, the OS rate was significantly higher in the CRT group than in the Non-CRT group (88% vs. 20% at 12-month; Figure 1D).” Moreover, we added Figure 1D, which shows the OS curves in 18 bladder cancer patients.
2. Authors suggest a single dose radiation would be sufficient in improving the combo therapy and thus set up two in vivo models to specifically irradiate the tumor in the left and not the right flank of the mice (methods). This is a key procedure, but it is not well depicted in the schematic in Figure 2A. It is more apparent in Figure 3A. As the radiation is site specific, but the other systemic therapies are not accurately depicting the schematics would improve the reader’s understanding of the experiments.
We appreciate the reviewer’s pointing out. We revised Figure 2A to describe that the left hindlimb was irradiated.
3. An additional point regarding these experiments, better description, depictions or presentation of the results in 2B and 3B would also help readers as well. The current labels are “irradiated” and “non-irradiated”; but the yellow and blue lines state they are in the CRT group that received radiation. With the brief legends and description in the results, it is difficult for me to understand if the right panels ever received irradiation. In addition, a direct comparison between the tumors in the left and right flanks of each mice would better illustrate the specific effects of radiation.
We appreciate the reviewer’s constructive comments. We changed the labels in Figure 2B/3B from “Irradiated tumors” and “Non-irradiated tumors” into “Irradiated tumors (left hindlimb)” and “Irradiated tumors (right flank)”.
Moreover, we revised figure legends in Figure 2A and B as follows (line 124-128): “A, Scheme for tumor inoculation and treatments. Six- to seven-week-old mice were inoculated subcutaneously with MB49 cells in the left hindlimb and in the right flank, and seven days later, received irradiation with a single fraction of 10 Gy to the left hindlimb and anti-PD-1 treatment (or its isotype controls). B, MB49 tumor growth curves of irradiated tumors in the left hindlimb (left) and non-irradiated tumors in the right flank (right).”
Similarly, we revised figure legends in Figure 3A and B as follows (line 181-189): “A, Scheme for tumor inoculation, treatments, and T cell analysis. For the CRT model (top), six- to seven-week-old mice were injected with MB49 cells in the left hindlimb, and seven days later, received cisplatin at 3 mg/kg and irradiation with a single fraction of 10 Gy to the left hindlimb. The mice were injected with MB49 cells in the right flank 14 days after irradiation, and seven days later, received anti-PD-1 treatment (or its isotype controls). For the Non-CRT model (bottom), only right flank tumor was established in mice, and cisplatin and irradiation were not given. B, MB49 tumor growth curves of irradiated tumors in the left hindlimb (left) and non-irradiated tumors in the right flank (right).”
Finally, to evaluate the specific effect of irradiation, we compared the growth rates of tumors in the left hindlimb between the control mice and the mice treated with irradiation alone and there was no significant difference between them (Figure 2B, left). Moreover, we directly compared the growth rates of irradiated tumors (left hindlimb) and non-irradiated tumors (right flank) of the mice treated with irradiation alone, and there was no significant difference between them. These results suggest poor anti-tumor effects of irradiation alone. Because the former result sufficiently illustrates the specific effect of irradiation, we did not comment the latter result in the revised manuscript. Thus, we added the following sentences to the Results section (line 112-115): “There was no significant difference of the growth rates of irradiated left hindlimb tumors between the mice treated with irradiation alone and the control mice (Figure 2B, left), reflecting poor anti-tumor effects of irradiation alone.”
4. Additional descriptions should be added in the results and potentially methods to justify why experiments were conducted in this particular mouse model/strain (C57BL/6J) and the cancer cell line (MB49). Particularly, emphasis should be placed on how the immune system of this model is suited towards studying urothelial cancer interaction with immune therapies. Additional citations would bolster the credibility of the results derived from these models.
We appreciate the reviewer’s helpful comments. A previous study used C57BL/6 mice inoculated with MB49 cells to investigate the radiation-induced abscopal effect in urothelial cancer (Rompré-Brodeur, A. et al. PD-1/PD-L1 Immune Checkpoint Inhibition with Radiation in Bladder Cancer: In Situ and Abscopal Effects. Mol Cancer Ther 2020, 19, (1), 211-220.). In their study, hypofractionated radiotherapy provoked abscopal effects in conjunction with concurrent anti-PD-L1 treatment. Based on this previous work, we evaluated whether abscopal effects can be induced by a single dose of irradiation in conjunction with concurrent anti-PD-1 treatment in C57BL/6J mice inoculated with MB49 cells.
Thus, we revised sentences in the Results section as follows (line 104-112): “A previous study showed that hypofractionated radiotherapy induced abscopal effects in conjunction with concurrent anti-PD-L1 treatment using C57BL/6 mice inoculated with murine UC cell line MB49 [15]. Based on this previous work, we conducted a pilot study of the MB49 model in which left hindlimb tumors were irradiated (right flank tumors were not irradiated) and either anti-PD-1 antibody or isotype control was administered (Figure 2A) to evaluate whether abscopal effects can be induced by a single dose of irradiation in conjunction with concurrent anti-PD-1 treatment in a UC mouse model.”